

# Transcriptome analysis and functional identification of GmMYB46 in soybean seedlings under salt stress

Xun Liu[1,2], Xinxia Yang[3] and Bin Zhang[1]

[1] College of Chemistry and Bioengineering, Hunan University of Science and Engineering, Yongzhou, China
[2] College of Life Sciences, Nanjing Agricultural University, Nanjing, China
[3] Department of Logistics, Hunan University of Science and Engineering, Yongzhou, China

## ABSTRACT

Salinity is one of the major abiotic stress that limits crop growth and productivity. We investigated the transcriptomes of salt-treated soybean seedlings versus a control using RNA-seq to better understand the molecular mechanisms of the soybean (*Glycine max* L.) response to salt stress. Transcriptome analysis revealed 1,235 differentially expressed genes (DEGs) under salt stress. Several important pathways and key candidate genes were identified by KEGG enrichment. A total of 116 differentially expressed transcription factors (TFs) were identified, and 17 TFs were found to belong to MYB families. Phylogenetic analysis revealed that these TFs may be involved in salt stress adaptation. Further analysis revealed that *GmMYB46* was up-regulated by salt and mannitol and was localized in the nucleus. The salt tolerance of transgenic *Arabidopsis* overexpressing GmMYB46 was significantly enhanced compared to wild-type (WT). GmMYB46 activates the expression of salt stress response genes (*P5CS1, SOD, POD, NCED3*) in *Arabidopsis* under salt stress, indicating that the GmMYB46 protein mediates the salt stress response through complex regulatory mechanisms. This study provides information with which to better understand the molecular mechanism of salt tolerance in soybeans and to genetically improve the crop.

## INTRODUCTION

Plant growth and development are often affected by unfavorable external stimuli such as drought, salinity, and extreme temperatures. Salt stress may lead to decreased crop yields and it reportedly affects approximately 6% of arable land and 20% of irrigated land (*Chen et al., 2021a*). Soil salinization affects plant growth differently at each developmental stage, typically through osmotic stress and ion toxicity. Osmotic stress is the primary stressor and immediately affects plant growth when the plant is exposed to high salinity. Ion toxicity occurs later when NaCl concentration has reached a threshold beyond which the ion homeostasis in plants is disrupted (*Van Zelm, Zhang & Testerink, 2020*). High salinity stress inhibits plants' nutrient uptake, thus inhibiting growth and development (*Van Zelm, Zhang & Testerink, 2020*). Salt stress stimulates the production of reactive oxygen species

Corresponding author
Bin Zhang,
zhangbin27104@huse.edu.cn

(ROS), such as hydrogen peroxide, superoxide, and hydroxyl radicals. ROS have a strong oxidative capacity and can cause cell plasma membrane damage, metabolic dysfunction, and cell death by disrupting redox homeostasis (*Liang et al., 2018*).

Plants have evolved many complex molecular and physiological mechanisms to adapt to salinized environments, such as the plant hormone signaling pathway, salt overly sensitive (SOS) signaling pathway, $Na^+$ efflux or sequestration in vacuoles, transcriptional regulation, improvement of antioxidant enzyme activity, and osmolyte (which include proline and soluble sugar) accumulation (*Van Zelm, Zhang & Testerink, 2020*). Protein kinases (PKs) and protein phosphatases (PPs) in plant signal transduction pathways participate in abiotic stress responses through protein dephosphorylation. The PP2C protein family is reportedly important to the salt stress response because they affect plants' metabolic process, hormone levels, and growth factors (*Wang et al., 2020a*). In addition, the maintenance of intracellular sodium and potassium homeostasis is essential for the survival of plants under salt stress, and many ion channels, transporters, and antiporters play a role in this process. For example, HKT1 (high-affinity potassium transporter 1) was initially thought to be an $Na^+/K^+$ symporter in wheat, and that $Na^+$ was excreted by transpiration to enhance plant salt tolerance (*Schachtman & Schroeder, 1994*). However, the *Arabidopsis* AtHKT1 protein selectively transports $Na^+$, affecting its distribution in roots and shoots (*Ren et al., 2005*). The tonoplast-localized NHX-type $Na^+/H^+$ exchangers have been reported to sequestrate $Na^+$ into vacuoles and participate in $K^+$ homeostasis. For example, plasma membrane-localized NHX7/SOS1 flushes excess $Na^+$ through the roots and restricts excessive accumulation, which regulates intracellular ion homeostasis and enhances plant salt tolerance. However, the *Arabidopsis sos1* mutant showed more sensitivity to salt stress (*Ji et al., 2013*; *Shi et al., 2002*). *Nieves-Cordones et al. (2010)* found that the high-affinity potassium transporter 5 (HAK5) could promote $K^+$ absorption under high salinity conditions, which was beneficial to maintaining the balance of $Na^+/K^+$. However, some HAK family members can regulate $Na^+$ translocation. *Wang et al. (2020b)* revealed that SlHAK20 transported $Na^+$ and $K^+$ and regulated $Na^+/K^+$ balance under salt stress. The mechanism for transcriptional regulation of plant salt tolerance has attracted much attention in the scientific community. Transcription factors (TFs) are considered to be gene switches, which activate or inhibit the expression of downstream stress-related genes by binding to specific cis-acting elements on promoters and forming a complex gene regulatory network to alleviate the damage caused by abiotic stress (*Hoang et al., 2017*). Many TFs are involved in regulating the response and adaptation of plants to salt stress, including WRKY (*Li et al., 2019a*), MYB (*Du et al., 2018*), AP2/ERF (*Zhao et al., 2019a*), NAC (*Li et al., 2021*). Plants respond to osmotic stress by accumulating a large number of osmolytes under high salinity conditions, these osmolytes generally include proline, sugar alcohols, sorbitol, and quaternary ammonium compounds (*Van Zelm, Zhang & Testerink, 2020*). *Liu et al. (2015)* found that the overexpression of *AtbHLH112* in *Arabidopsis* activated the expression of *P5CS* and inhibited the expressions of the *P5CDH* and *ProDH* genes, thus increasing proline synthesis while reducing its degradation. The increased proline content contributes to better abiotic stress tolerance of *Arabidopsis*.

Soybean (*Glycine max* L.) is a global staple edible oil and food crop, which also provides the raw material for the manufacture of animal feeds. Salt stress adversely affects the growth of soybean plants, which leads to reduced yields. At present, many key genes related to salt tolerance have been identified in soybeans. The CHX-type ion transport protein GmSALT3 improves the salt tolerance of soybean plants by promoting $Na^+$ and $Cl^-$ exclusion (*Qu et al., 2021*). The salt tolerance of soybean hairy roots and transgenic *Arabidopsis* overexpressing *GmCHX20a* decreases, promoting the absorption and accumulation of $Na^+$ in roots be due to GmCHX20a, while GmCHX1 (GmSALT3) was shown to protect transgenic *Arabidopsis* plants *via* $Na^+$ exclusion under salt treatments (*Jia et al., 2021*). The expression level of the MYB transcription factor *GmMYB84* was up-regulated under salt stress, and the overexpression of *GmMYB84* enhanced the salt tolerance of transgenic soybeans (*Zhang et al., 2020a*). Salt-induced soybean NAC transcription factor, GmSIN1, promoted root growth and enhanced salt tolerance by controlling abscisic acid production and ROS (*Li et al., 2019b*). However, as there are limited genetic resources to improve soybean genetics these require further study.

Understanding the molecular mechanisms of plant salt tolerance is a prerequisite for the development of salt-tolerant crop varieties, and technological advances can help to clarify the complex mechanisms of plant response to salt stress. Transcriptome analysis is a powerful tool to uncover stress-related regulatory networks in plants. It is widely used to identify stress-responsive genes and genetic control elements. A large number of salt-induced genes have been identified in oilseed rape (*Yong et al., 2014*), rice (*Zhou et al., 2016*), watermelon (*Song, Joshi & Joshi, 2020a*), kenaf (*Munsif et al., 2020*), and maize (*Chen et al., 2021b*). *Ali et al. (2012)* found that soybeans responded to salt stress through abscisic acid (ABA) biosynthesis based on tag sequence data combined with GO analysis. *Zhao et al. (2019a)* identified differentially expressed genes through transcriptome analysis, including *Glyma.02G228100*, *Glyma.03G226000*, *Glyma.03G031000*, *Glyma.03G031400*, *Glyma.04G180300*, *Glyma.04G180400*, *Glyma.05G204600*, *Glyma.08G189600*, *Glyma.13G042200*, and *Glyma.17G17320*, which were considered to be the key genes involved in the salt tolerance mechanism in salt-tolerant soybeans. It is generally believed that plants sense salt stress mainly through their roots, but high salinity may lead to physiological changes such as growth inhibition, leaf wilting, excessive accumulation of ROS in leaves, and increased $Na^+$ content. However, the molecular mechanism of plant shoots (especially leaves) in response to salt stress is not well-established. In this study, the shoot (including stems and leaves) of soybean seedlings was used for transcriptome sequencing (RNA sequencing, RNA-seq), and Kyoto Encyclopedia of Genes and Genomes (KEGG) enrichment combined with other bioinformatics methods were used to identify key pathways and genes induced by salt stress. This study will provide data for the investigation of molecular mechanisms of the salt response in soybeans and cultivation of salt-resistant varieties.

## MATERIALS AND METHODS

### Plant growth and salt treatment

Cultivated soybean seeds (Tianlong No. 1) were sterilized with 75% ethanol for 10 min, washed with sterile distilled water, and laid on wet filter paper for two days. The germinated seeds were transferred to quartz sand and irrigated with Hoagland nutrient solution, then cultured in a plant growth chamber (16 h day/8 h night, $25 \pm 2$ °C, 60–70% relative humidity). After 10 d, soybean seedlings with consistent growth were selected and divided into two groups: seedlings irrigated with Hoagland nutrient solution as control, and those irrigated with Hoagland nutrient solution containing 120 mM NaCl as the treatment. The shoots (stems and leaves) were obtained after being treated for 6 h and were frozen in liquid nitrogen and stored at $-80$ °C for transcriptome sequencing. The control group and treatment group each contained three biological replicates.

### Phenotypic comparison and physiological measurements

Histochemical staining with NBT (Nitrotetrazolium blue chloride) and DAB (3, 3-Diaminobenzidine) was used to examine the ROS ($O_2^-$ and $H_2O_2$) accumulation levels in the leaves (*He et al., 2019*). After 0 h, 3 h, 6 h, and 12 h salt treatment, the leaves were removed and immersed in DAB and NBT staining solution. Chlorophyll was removed with alcohol for photography after 12 h at room temperature. After 6 days of salt treatment, plant phenotypes were analyzed and compared, and physiological indexes related to salt stress (including plant height, fresh weight per plant, proline content, $Na^+$, and $K^+$ ion content) were determined as previously reported (*Zhang et al., 2020b*).

### RNA isolation, cDNA library preparation, and Illumina sequencing

The total RNA was isolated from soybean shoot samples using a mirVana miRNA Isolation Kit (Invitrogen, AM1561) according to the manufacturer's protocol. The contained DNA was digested with DNase I. The quality and concentration of RNA was detected using Agarose gel electrophoresis and Nanodrop 2000 (ThermoFisher Scientific, Massachusetts, USA), respectively. The mRNA in the total RNA was enriched using magnetic beads with Oligo (dT), and approximately 4 µg mRNA was collected from each sample for library construction using a TruSeq Stranded mRNA LTSample Prep Kit (Illumina, San Diego, USA) following the manufacturer's protocol. The libraries were then sequenced on the Illumina sequencing platform (Illumina HiSeqTM 2500). The raw reads generated in this study were deposited in the NCBI database under accession number PRJNA741583.

### Validation by qRT-PCR analysis

The expression of 10 randomly selected genes was evaluated by qRT-PCR analysis to validate the transcriptome data. This selection included 5 up-regulated genes (*LOC100807235, LOC100816551, LOC100785783, LOC100787314, LOC100795929*) and 5 down-regulated genes (*LOC100805378, LOC102663255, LOC100306125, LOC100787705, LOC100819491*). qRT-PCR was conducted using Bio-Rad CFX96 PCR System (USA, Bio-Rad) with a 20 µL reaction mixture. The mixture consisted of 10 µL HieffTM qPCR SYBR® Green Master Mix (No Rox Plus) (11201ES, Yeasen, Shanghai, China), 0.5 µL (10 µM) Primer F, 0.5 µL

(10 μM) Primer R, and 9 μL (100 ng/μL) diluted cDNA. Relative expression of the genes was analyzed with the $2^{-\Delta\Delta Ct}$ method using the *GmEF-1α* gene as an internal control to normalize the level of gene expression (*Zhao et al., 2020*). The primers for qRT-PCR analysis were designed using the Primer Blast tool on the NCBI (http://www.ncbi.nlm.nih.gov/). All primers used in this study are shown in Table S1.

## RNA-Seq analysis

The transcriptome sequencing was conducted by OE Biotech Co., Ltd. (Shanghai, China). Raw reads containing ploy-N and the low-quality reads were filtered to obtain the clean reads using Trimmomatic software (*Bolger, Lohse & Usadel, 2014*). The clean reads were mapped to the cultivated soybean (*Glycine max* (Linn.) Merr.) genome (https://www.ncbi.nlm.nih.gov/genome/5?genome_assembly_id=401179). The Q20, Q30, and GC content from the clean read data were calculated and used to evaluate all samples' sequencing quality. All further analyses were based on high-quality data from clean reads. Pearson correlation analysis was performed to identify the repeatability of three biological replicates in the same group.

The gene expression levels were normalized to fragments per kilobase of transcript per million fragments mapped (FPKM) using Cufflinks v2.2.2 software. Differentially expressed genes (DEGs) were identified by comparing the read counts of the control and salt-treated samples' gene transcripts using the DESeq R package function's estimate size factors and nbinom tests (*Anders & Huber, 2010*). The $p$-value < 0.05 and |log2 fold change| > 1 or $p$-value < 0.01 and |log2 fold change| > 1.5 were set as the threshold for significantly differential expression (*Long et al., 2019*).

KEGG pathways' enrichment of DEGs was determined using KOBAS software with FDR < 0.01. All of the genes annotated as transcription factors in DEGs were screened to identify the response of transcription factors to salt stress in soybean shoots.

## Phylogenetic analysis of MYB family members

Some members of the MYB families reportedly involved in plant abiotic stress were retrieved from the NCBI database, and phylogenetic analysis was conducted with the corresponding family members in the salt-treated soybeans' DEGs. The phylogenetic tree was constructed using the neighbor-joining (NJ) method with 1,000 bootstrap replicates in MEGA X (*Zhang et al., 2018*). The FPKM values of 17 differentially expressed MYBs were used to draw the heatmap using TBtools (v1.0983) (*Chen et al., 2020*).

## Expression pattern and subcellular localization of GmMYB46

Ten-day old soybean seedlings were treated with 120 mM NaCl or 300 mM Mannitol. The leaves were removed at 0, 3, 6, and 12 h, quick-frozen with liquid nitrogen, and stored at −80 °C for RNA extraction and detection of *GmMYB46* expression pattern using semi-quantitative PCR. The CDS removed the stop codon of *GmMYB46,* which was cloned and ligated into the pCAMBIA1300-GFP vector to obtain a fusion expression vector *GmMYB46*:eGFP. This vector was introduced into *Agrobacterium tumefaciens* GV3101 before the positive *A. tumefaciens* was injected into *Nicotiana benthamiana* leaves. Plants

were cultured for 2 d. The GFP fluorescence was observed and imaged using a laser confocal microscope (Zeiss LSM880 Meta, Jena, Germany).

## Salt tolerance analysis of transgenic *Arabidopsis*

The full-length CDS sequence of *GmMYB46* was cloned to construct the overexpressed vector 35S:*GmMYB46* and the wild-type *Arabidopsis* Col-0 was transformed using the floral dip method by *Agrobacterium*-mediated. The transgenic lines were identified by hygromycin resistance screening and semi-quantitative PCR. The seeds were evenly sown on nutrient-rich soil, kept in the dark at 4 °C for 3 d, and then placed in a growth chamber for 10 d. The control group (CK) was irrigated with deionized water for culturing, and the treatment group (NaCl) was irrigated with 150 mM NaCl solution. The phenotype was observed and photographed after 5 d and 7 d of treatment, and physiological indicators (survival rate, relative water content, malondialdehyde (MDA) content, proline content, SOD, and POD activities) were measured on day seven, following previously reported methods (*Chen et al., 2019*; *He et al., 2019*). The leaves were removed for RNA extraction 12 h after salt treatment, qRT-PCR analysis of salt stress-related genes was performed, and *AtACTIN2* was used as an internal reference using the same methods as noted above.

## Statistical analysis

All data were statistically analyzed using IBM SPSS Statistics software (v20.0, NY, USA). Means $\pm$ SD are shown in this study ($n \geq 3$). The Student's $t$-test was used to calculate $P$-values ($P < 0.05$) to determine the significance of the differences among treatments.

# RESULTS

## Salt injury in soybean shoots exposed to high salinity

To observe the effect of salt stress on the growth of soybean shoots, 10-day-old soybean seedlings were selected for salt treatment. NBT and DAB staining results showed that the contents of $H_2O_2$ and $O_2^-$ in leaves were significantly higher than those at 0 h and 3 h after salt treatment for 6 h and 12 h (Fig. 1A), indicating that excessive ROS had been generated and accumulated in plant leaves at 6 h salt treatment. Therefore, the 6 h salt-treated plants were used for transcriptome sequencing. The phenotypic comparison showed that the growth of soybean seedlings after 6 d of salt treatment was significantly inhibited, and plant height and fresh weight were lower than those of the control (CK) (Figs. 1B, 1C, 1D). In addition, the content of $Na^+$ in shoots of salt-treated soybean seedlings increased significantly and $K^+$ decreased compared with CK (Figs. 1E, 1F). Compared with CK, a large amount of proline accumulated in the shoot of the salt-treated soybean (Fig. 1G).

## Transcriptome analyses

RNA-Seq analysis was performed on seedlings treated with 0 mM (CK) and 120 mM NaCl for 6 h. We constructed six libraries from the control and NaCl treatment groups (three biological replicates in each group) for analysis using RNA-Seq. These libraries yielded 50.05 to 58.13 million raw reads, and 48.85 to 56.88 million clean reads (Table 1). The Q30 in each library was greater than 92%, indicating that the quality of the sequencing data

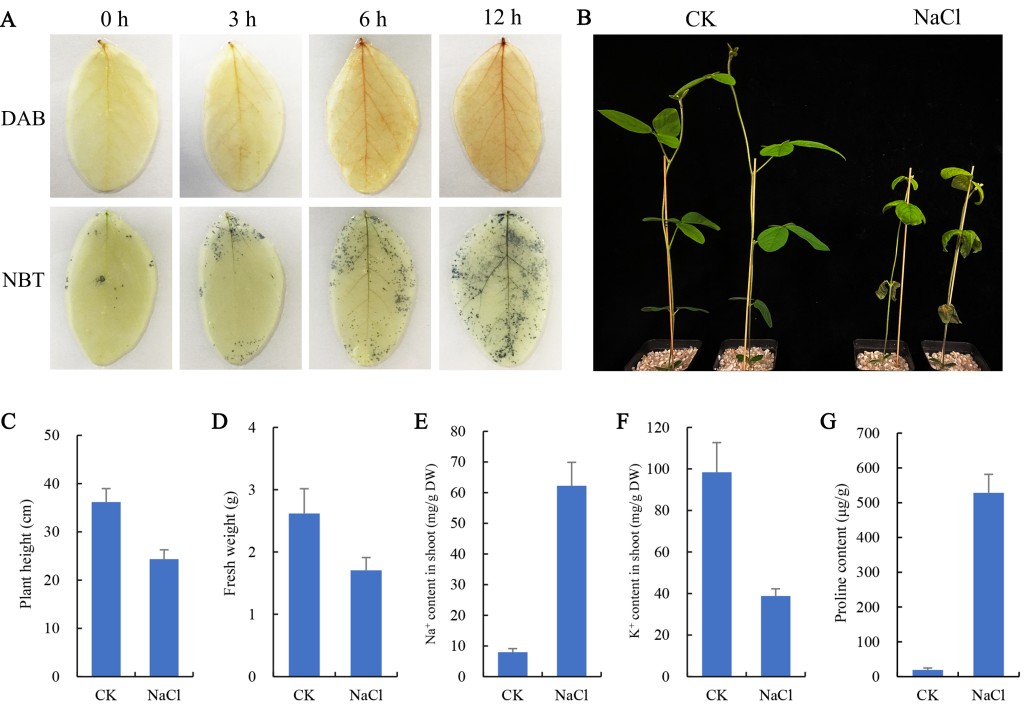

**Figure 1  Effects of salt stress on growth and physiological indicators of soybean seedlings.** (A) After salt treatment (0, 3, 6, 12 h), the accumulation of $O_2^-$ and $H_2O_2$ in leaves was detected by NBT and DAB staining, respectively, (B) phenotypic changes of soybean seedlings after treatment with 100 mM NaCl for 6 days, (C) plant height, (D) fresh weight, (E) $Na^+$ content and (F) $K^+$ content in shoots, and (G) proline content under normal (CK) and salt treatment conditions. Data represent mean ± SD ($n = 6$), *$P < 0.05$ (Student's t test).

could be used for further analysis. Approximately 98% of the clean reads were mapped to the soybean genome. Correlation analysis revealed that the relationships among the three biological replicates in the same group were more closely related (Fig. 2A). The volcano plot showed that 466 genes were up-regulated and 769 genes were down-regulated after salt treatment based on the loose DEGs restrictions ($p < 0.05$, |Log2 fold change| > 1) (Fig. 2B, Table S2). However, stricter parameters revealed ($p < 0.01$, |Log2 fold change| > 1.5) 425 DEGs, including 127 up-regulated and 298 down-regulated genes (Table S3). Transcriptome data was verified by qRT-PCR analysis and showed that the fold variation between RNA-seq expression and qRT-PCR analyses were linearly correlated ($R^2 = 0.8568$) (Fig. 2C).

KEGG enrichment analysis of up- and down-regulated DEGs was conducted using the KOBAS database. The top 20 most highly enriched pathways were displayed (Fig. 3). The up-regulated DEGs induced by salt treatment were significantly enriched in 'Plant hormone signal transduction' (34), 'Phenylalanine metabolism' (4), 'Fructose and mannose metabolism' (6), 'Carotenoid biosynthesis' (4), and 'MAPK signaling pathway' (8) (Fig. 3A). The down-regulated DEGs were significantly enriched in 'Plant hormone signal transduction' (25), 'Ribosome biogenesis in eukaryotes' (9), 'MAPK signaling

**Table 1  Original RNA-seq data and quality control analysis.**

| Sample | Raw reads (M) | Clean reads (M) | Q30 (%) | Total reads | Total mapped reads |
|--------|---------------|-----------------|---------|-------------|--------------------|
| CK-1   | 50.49 | 49.3  | 93.26 | 49300196 | 48248935 (97.87%) |
| CK-2   | 50.05 | 48.85 | 93.52 | 48851760 | 47818910 (97.89%) |
| CK-3   | 58.13 | 56.88 | 92.85 | 56884250 | 55613843 (97.77%) |
| NaCl-1 | 52.01 | 50.88 | 93.04 | 50878700 | 49752466 (97.79%) |
| NaCl-2 | 50.2  | 49.11 | 93.02 | 49105610 | 48029838 (97.81%) |
| NaCl-3 | 56.17 | 54.94 | 92.89 | 54943854 | 53713686 (97.76%) |
| Total  | 317.05 | 309.96 |       |          |                   |

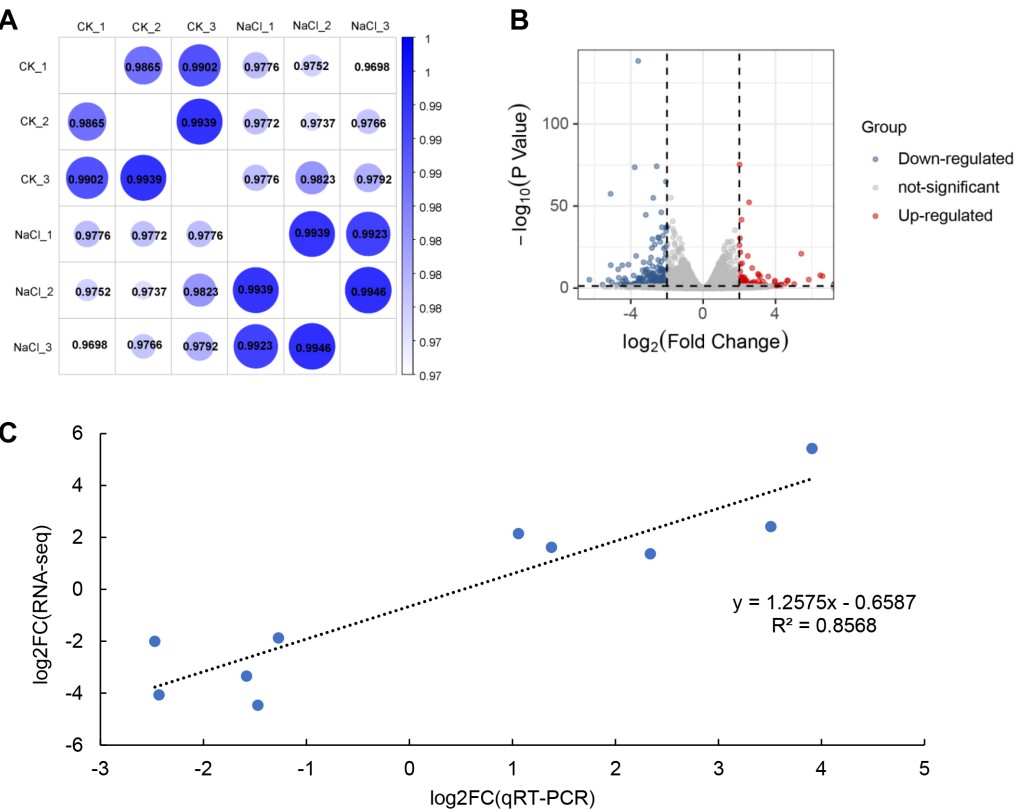

**Figure 2  RNA-seq analysis of soybean shoots under salt stress.** (A) Correlation analysis of control (CK) and salt-treated samples (NaCl), (B) volcano plot of DEGs. Red dots indicate up-regulated genes and blue dots indicate down-regulated genes. (C) Correlations in qRT-PCR ($x$-axis) RNA-seq data ($y$-axis) using 10 randomly selected DEGs.

pathway' (13), 'Plant-pathogen interaction' (15), and 'Arginine and proline metabolism' (6) (Fig. 3B).

## Key genes in early response to salt stress

The up- and down-regulated DEGs can be divided into the following five categories in the KEGG database: Cellular Processes, Environmental Information Processing, Genetic Information Processing, Metabolism, and Organismal Systems. Among them, the number

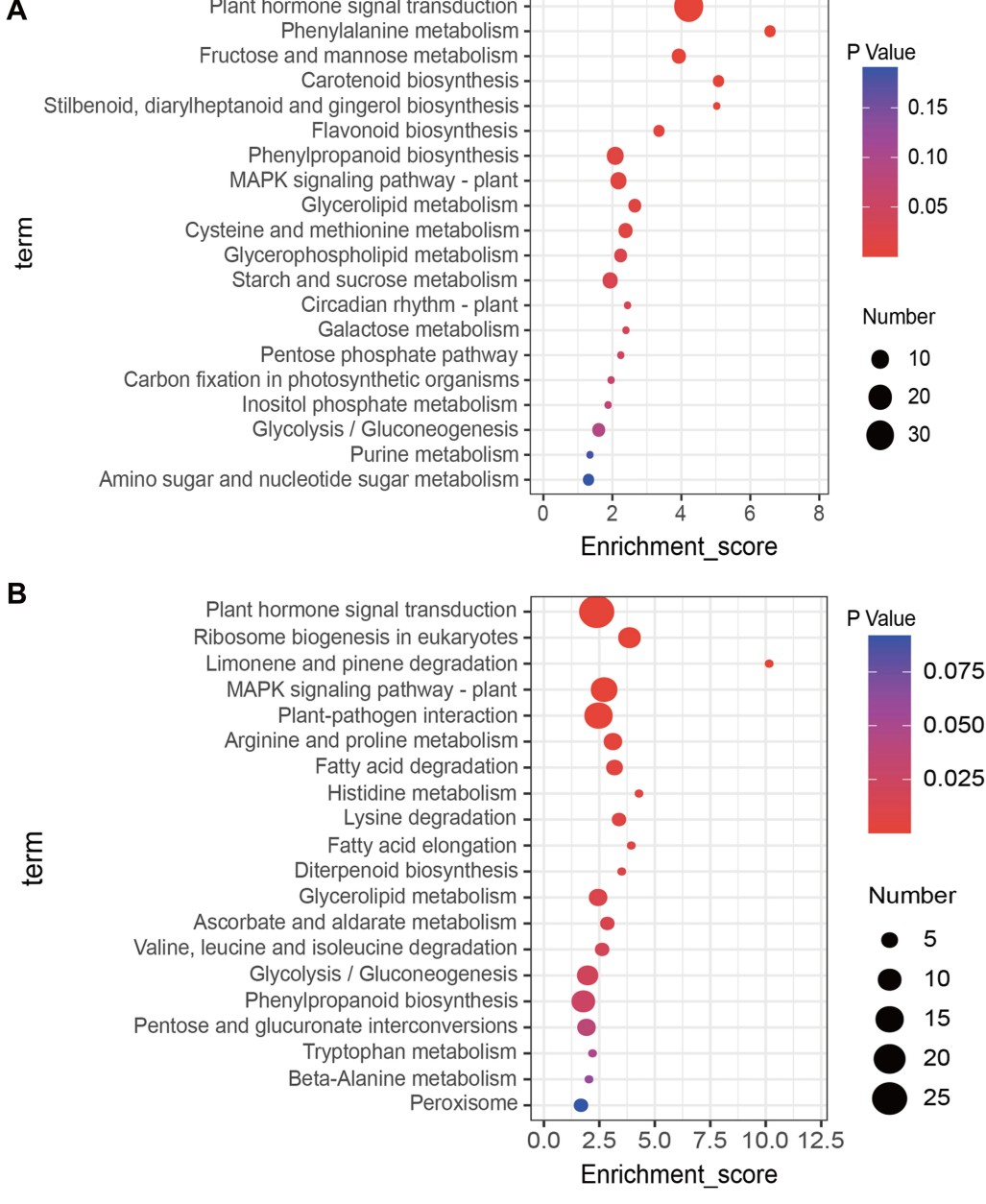

**Figure 3** Kyoto Encyclopedia of Genes and Genomes (KEGG) analysis of DEGs in response to salt stress using the top twenty significant enrichment KEGG terms. (A) KEGG terms enriched by up-regulated DEGs, (B) KEGG terms enriched by down-regulated DEGs.

of up-regulated and down-regulated DEGs is the largest in signal transduction categories, which belong to Environmental Information Processing, and include 34 and 29 genes, respectively (Table 2). The 34 up-regulated genes contain hormone-related proteins and PP2C family members, while down-regulated genes mainly include the ABA-related genes and probable xyloglucan endotransglucosylase/hydrolase protein family members

**Table 2  Classification statistics for DEGs of up-regulated (UR) and down-regulated genes (DR) in salt-stressed soybean according to KEGG pathway analysis.**

| KEGG categories | Number of UR | Number of DR |
|---|---|---|
| **Cellular Processes** | | |
|    Transport and catabolism | 1 | 10 |
| **Environmental Information Processing** | | |
|    Signal transduction | 34 | 29 |
|    Membrane transport | 1 | 0 |
| **Genetic Information Processing** | | |
|    Translation | 1 | 14 |
|    Folding, sorting and degradation | 5 | 9 |
|    Transcription | 0 | 5 |
|    Replication and repair | 0 | 3 |
| **Metabolism** | | |
|    Nucleotide metabolism | 3 | 3 |
|    Metabolism of terpenoids and Polyketides | 7 | 11 |
|    Metabolism of other amino acids | 2 | 8 |
|    Metabolism of cofactors and vitamins | 2 | 4 |
|    Lipid metabolism | 10 | 19 |
|    Energy metabolism | 4 | 3 |
|    Carbohydrate metabolism | 28 | 20 |
|    Biosynthesis of other secondary Metabolites | 12 | 13 |
|    Amino acid metabolism | 16 | 16 |
| **Organismal Systems** | | |
|    Environmental adaptation | 4 | 15 |

(Table S4). Environmental adaptation categories include four up-regulated genes and 15 down-regulated genes (Table 2, Table S5). Furthermore, seven genes from the ABA receptors PYL family enriched in both the plant hormone signal transduction pathway and the MAPK signaling pathway were significantly down-regulated, including *LOC100788576, LOC100819216, LOC100804400, LOC100805378, LOC100792229, LOC100798803*, and *LOC100788199* (Figs. 4A, 4E). The *PP2C* (Phosphatase 2C) family genes downstream of PYL were significantly up-regulated and include *LOC100793293, LOC100803764, LOC100807235, LOC100818568, LOC100776356, LOC100796161*, and *LOC100781388* (Figs. 4B, 4F). The expression levels of ABF family members downstream of SNF1-related protein kinase 2 (*SNRK2*), *LOC100819313* and *GmbZIP1*, were significantly up-regulated (Fig. 4A). Proline dehydrogenase genes (*PDH, LOC100797464*, and *LOC100799876*) in the arginine and proline metabolic pathways were significantly down-regulated (Figs. 4C, 4G) The phenylalanine ammonia-lyase genes (*GmPAL1.2, GmPAL1.3, GmPAL2.2*, and *GmPAL2.4*) in the phenylalanine metabolism pathway were significantly up-regulated to promote the production of cinnamic acid (Figs. 4D, 4H), which is important for salt stress adaptation.

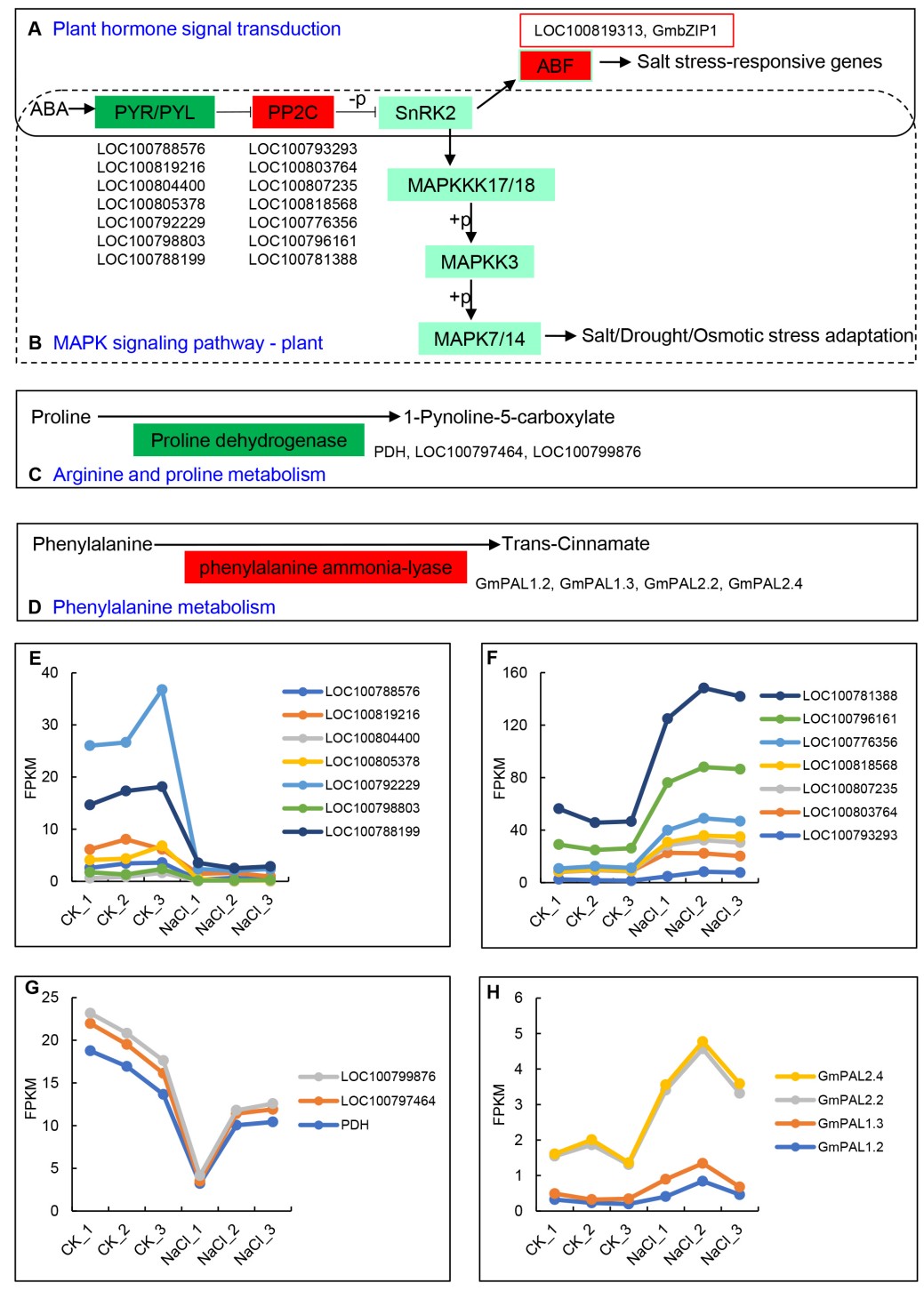

**Figure 4** **Key KEGG pathways of related genes in response to salt stress and salt-induced expression patterns.** (A) Plant hormone signal transduction, (B) MAPK signaling pathway, (C) arginine and proline metabolism, (D) phenylalanine metabolism, and (E–H) *PYR/PYLs*, *PP2Cs*, proline dehydrogenase genes, and phenylalanine ammonia-lyase gene expression represented using the FPKM value.

## Transcription factors (TFs) involved in salt stress

The differentially expressed TFs in DEGs were statistically analyzed, as shown in Fig. 5. We found that 1,235 DEGs contained a total of 116 transcription factor coding genes, accounting for 9.4% of DEGs. Of these, 49 were up-regulated and 67 were down-regulated. There were a large number of AP2/ERF, MYB, and bHLH families with 32, 17, and 14 differentially expressed TFs, respectively. Further analysis showed that the MYB family had the most members (11) in the up-regulated TFs, while the AP2/ERF family had the most members (26) in the down-regulated TFs. We searched MYB proteins that were reportedly involved in plant abiotic stress and conducted the phylogenetic analysis with MYB members in soybean DEGs. As shown in Fig. 6A, the up-regulated *LOC778071* and *SlMYB102* (*Solyc02g079280*) (*Zhang et al., 2020c*) were highly homologous and in the same clade; *LOC100799410*, *MdMYB46* ( *XP_008363629*) (*Chen et al., 2019*), and *TaMYB86B* (*KM066946*) (*Song, Joshi & Joshi, 2020a*) were in the same clade, and *LOC100805341* and *OsMYB2* (*AK120551*) (*Yang, Dai & Zhang, 2012*) were in the same clade. Further analysis showed that the expression level of *LOC100799410* was the most up-regulated by salt stress (fold change = 7.49) (Fig. 6B). Therefore, for further functional exploration *LOC100799410* was named *GmMYB46* according to the results of phylogenetic analysis.

## *GmMYB46* was up-regulated by salt treatment and localized in the nucleus

Semi-quantitative PCR analysis showed that the expression level of *GmMYB46* was significantly increased in leaves treated with salt for 6 and 12 h, while the expression of *GmMYB46* first increased (6 h) and then decreased (12 h) under mannitol treatment (Fig. 6C). In addition, the GFP fusion vector *GmMYB46*:eGFP was constructed and the transient expression was carried out in tobacco leaves. The results showed that green fluorescence was only observed in the nucleus of cells of *GmMYB46*:eGFP transformed leaves, indicating that GmMYB46 protein was localized in the nucleus (Fig. 6D).

## Overexpression of *GmMYB46* enhances the salt tolerance of transgenic *Arabidopsis*

An overexpression vector (35S:*GmMYB46*) was constructed to explore the function of *GmMYB46* in plant salt tolerance, and transgenic *Arabidopsis* lines (OX46.1, OX46.2) were obtained (Figs. S1A, S1B). Phenotypic analysis showed that the growth of *Arabidopsis* seedlings was significantly inhibited by 150 mM NaCl treatment for 5 d and 7 d, but OX46.1 and OX46.2 plants were less affected by salt than WT (Col-0) (Fig. 7A). The survival rates of OX46.1 and OX46.2 were significantly higher than those of the WT after 5 and 7 d of salt treatment (Fig. 7B). In addition, the relative water content of OX46.1 and OX46.2 plants was significantly higher than that of WT (Fig. 7C), while the MDA content was significantly lower than that of WT under salt stress (Fig. 7D). The proline content and the activities of SOD and POD increased significantly, while the overexpressed lines were significantly higher than that of WT (Figs. 7E, 7F). In addition, we examined the expression of salt stress-related genes in normal and salt-treated *Arabidopsis*. The results showed that the expression level of *P5CS1*, a key gene for proline synthesis, was significantly up-regulated in Col-0, OX46.1, and OX46.2 plants induced by salt, and it was significantly higher in

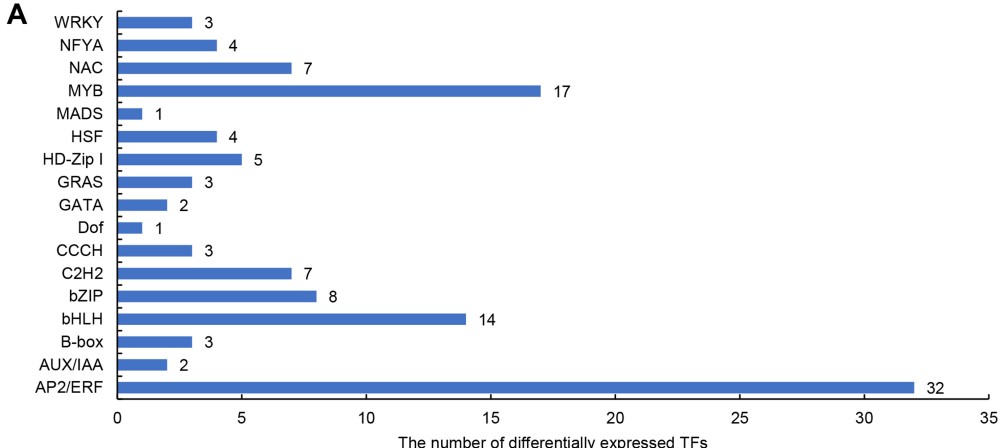

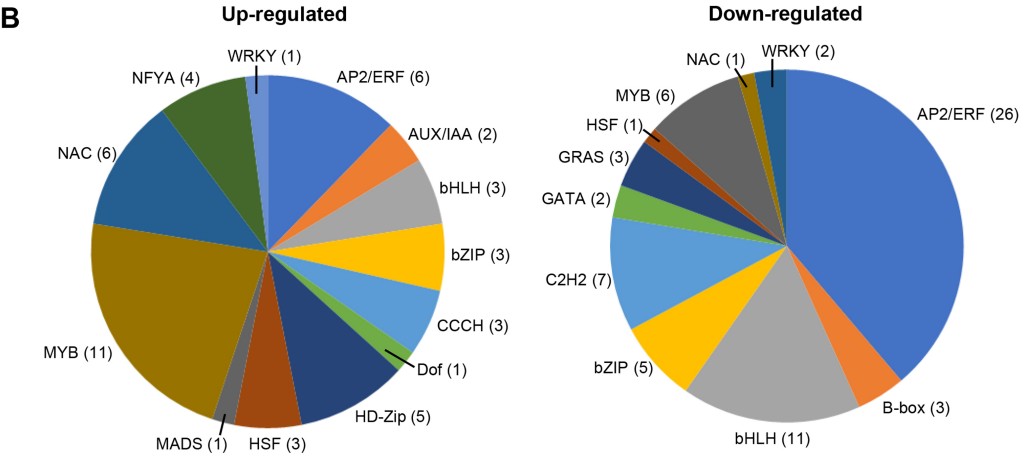

**Figure 5** **Differentially expressed TFs induced by salt stress in soybean.** (A) The classification and number of differentially expressed TFs, (B) classification of up-regulated and down-regulated TFs.

overexpressed lines than in WT (Fig. 8A). Antioxidant enzyme synthesis-related genes *SOD*, *POD*, and *NCED3*, a key gene for ABA biosynthesis, also showed a salt-induced expression pattern similar to *P5CS1* (Figs. 8B, 8C, 8D).

## DISCUSSION

Salt stress is one of the major environmental factors limiting plant growth and productivity. Plants have evolved complex physiological, biochemical, and molecular regulatory mechanisms to adapt to adverse conditions. These involve the expression changes of a large number of regulatory genes. Transcriptome sequencing is an effective tool for comprehensive analysis at the transcriptional level for plants under abiotic stress (*Zhang et al., 2020b*). There have been many studies on the transcriptome analysis of legumes using RNA-seq, including medicago (*Shu et al., 2018*), glycine (*Sun et al., 2016*), common bean (*Hiz et al., 2014*), and *Arachis hypogaea* (*Zhang et al., 2020b*). *Sun et al. (2016)* performed

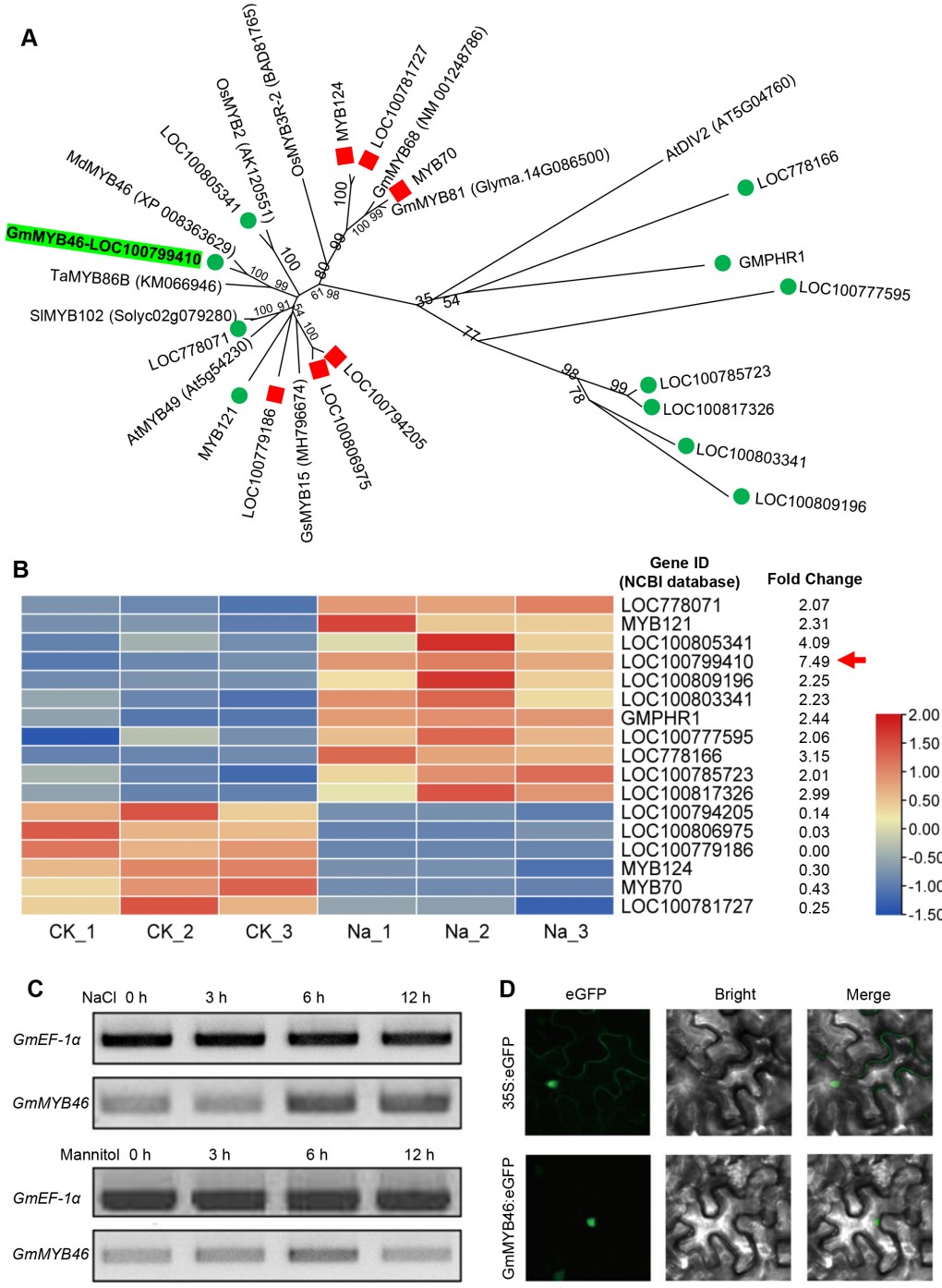

**Figure 6 Phylogenetic analysis of differentially expressed MYB TFs, and expression pattern and subcellular localization of GmMYB46.** (A) The phylogenetic tree of differentially expressed MYBs and other reported MYBs involved in abiotic stress, (B) heatmap diagrams showing the relative expression levels of differentially expressed MYBs based on the FPKM values, (C) expression analysis of *GmMYB46* induced by NaCl and mannitol, (D) subcellular localization of GmMYB46 in tobacco leaves.

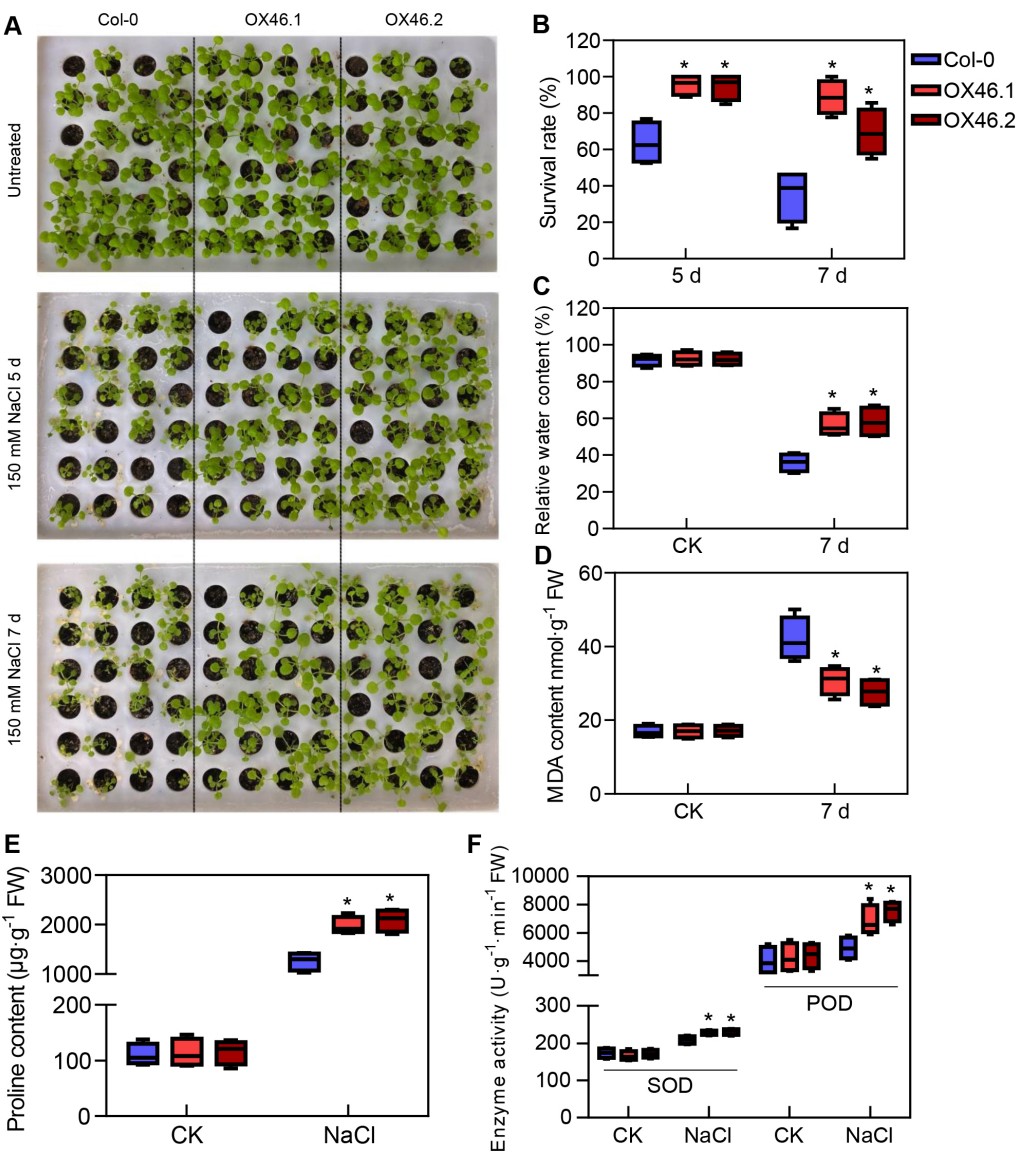

**Figure 7 Overexpression of *GmMYB46* enhanced the salt tolerance of transgenic *Arabidopsis*.** (A) Phenotypic analysis of wide-type (Col-0) and overexpressed lines (OX46.1 and OX46.2), (B) survival rate of Arabidopsis seedlings, (C) relative water content, (D) MDA content, (E) proline content, and (F) enzyme activities of SOD and POD. The data represent the mean ± SD. *$P < 0.05$ (Student's t test).

small RNA transcriptome analysis on root tip tissues of normal and salt-treated soybean seedlings and identified 66 salt-responsive miRNAs, and then demonstrated that salt-induced miR399 played an important role in regulating the developmental plasticity of soybean roots. Plant roots are known to directly encounter salinized soil as the first line of defense; accordingly, most studies have focused on the molecular regulatory mechanisms of the root response to salt stress (*Zhang et al., 2016*; *Zhao et al., 2020*). However, these molecular mechanisms and regulatory networks have not been studied in plant shoots when

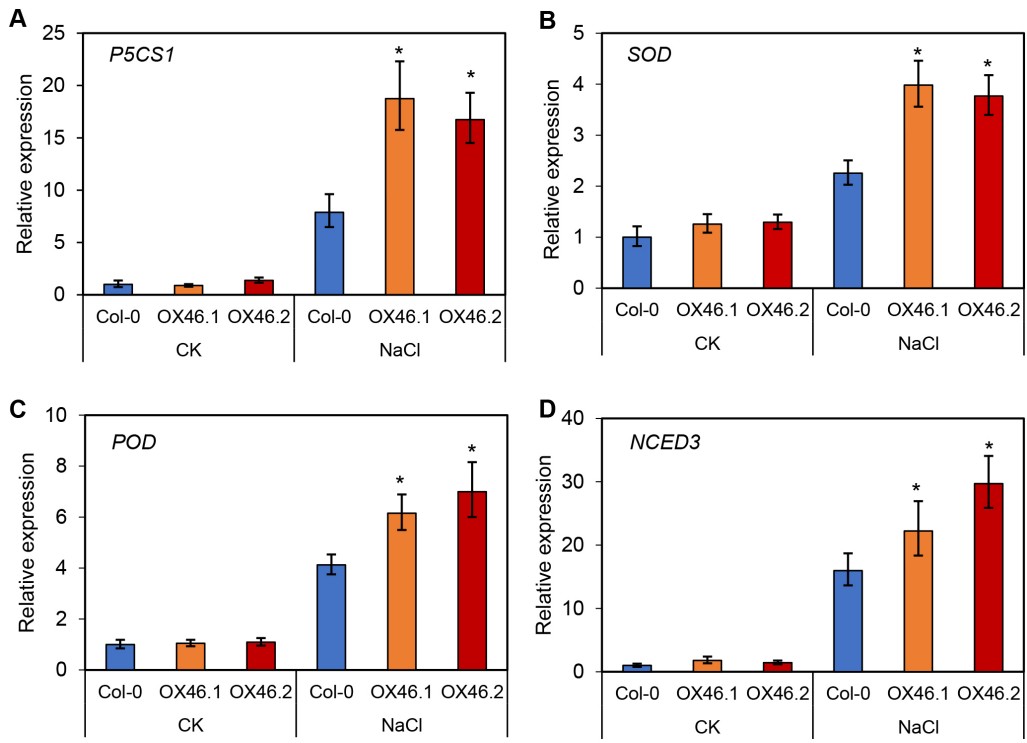

**Figure 8** **Expression patterns of stress-responsive genes in Col-0 and overexpressed lines (OX46.1 and OX46.2) in response to salt stress.** The expression of (A) *P5CS1*, (B) *SOD*, (C) *POD*, and (D) *NCED3* genes.

growth is inhibited due to salt stress. *Zeng et al. (2019)* performed transcriptome sequencing on salt-treated soybean leaves, however only eight genes were preliminarily identified, and the function of these genes has not been explored in plants in their report. In contrast to previous work, shoot tissues (including stems and leaves) of salt-treated soybean seedlings were used for transcriptome sequencing in this study, which may better reflect the response of the above-ground parts of soybeans to salt stress. More importantly, transcriptome and bioinformatics analysis combined with *Arabidopsis* genetic transformation technique were used to preliminarily demonstrate the function of *GmMYB46* in response to salt stress in our work.

We identified some important pathways and key candidate genes that may be involved in the salt stress response among 1,235 DEGs. *Wu et al. (2017)* conducted KEGG enrichment analysis of salt-induced DEGs and found 'Plant hormone signal transduction', 'MAPK signaling pathway', 'Arginine and proline metabolism' and other secondary metabolite pathways were significantly enriched. *Tian et al. (2018)* used KEGG analysis to determine that 'Plant hormone signal transduction pathway' was significantly enriched in both salt-sensitive and salt-tolerant *Rosa chinensis*. Similarly, our analysis found that DEGs were significantly enriched in 'Plant hormone signal transduction', 'MAPK signaling pathway', 'Phenylalanine metabolism', 'Arginine and proline Metabolism' pathways (Fig. 3). These pathways play an important role in the process of plant salt stress response and the results

provide a reference for salt tolerance genes in soybeans and assist in analyzing molecular regulatory mechanisms.

ABA is one of the most important salt stress hormones. It plays a crucial role in salt stress defense. ABA is involved in the regulation of osmotic stress, ion homeostasis, and the scavenging of reactive oxygen species induced by salt injury (*Yu et al., 2020*). It is widely believed that PYR/PYL protein, PP2Cs, and SnRK2 kinase constitute the core module of ABA signaling (PYL-PP2CS-SnRK2s), which is responsible for the earliest ABA signal recognition and transduction in plants (*Gong et al., 2020*). *Liu et al. (2012)* showed that overexpression of *AtPP2CG1* enhanced the salt tolerance of transgenic *Arabidopsis* by activating the expression of salt stress response genes *RD29A, RD29B, DREB2A,* and *KIN1*. In this study, we found that the ABA receptor PYL family genes significantly enriched in 'Plant hormone signal transduction' and 'MAPK signaling pathway' were significantly down-regulated, while PP2C family members were significantly up-regulated. Two downstream ABF family genes (*LOC100819313* and *GmbZIP1*) were also significantly up-regulated (Fig. 4). *Lei et al. (2021)* conducted transcriptome sequencing of salt-treated *Ricinus communis* L. and found that *PP2C* genes were up-regulated in salt-treated cultivated *R. communis*, which inhibited the activity of SnRK2 kinase and further promoted the up-regulation of downstream ABF genes. The results in this work are consistent with previous reports, further confirming the function of the ABA signaling pathway PYL-PP2CS-SnRK2s in short-term salt stress. In addition, previous reports have shown that proline metabolism is an important pathway in response to oxidative stress caused by salt injury, and that the accumulation of proline is protective for plants (*Verbruggen & Hermans, 2008*). In this study, three key genes of the proline degradation pathway (*PDH, LOC100797464, LOC100799876*) were significantly down-regulated (Fig. 4), which reduce the degradation of proline in shoots and increase its accumulation in response to salt stress. Phenylalanine ammonia-lyase (PAL) is a key enzyme in the phenylalanine pathway that catalyzes the deamination of phenylalanine to produce cinnamic acid. Some *PAL* genes (*Aradu.IU1HH, Araip. 69J63, Araip.GM19P,* and *Araip.V9S7Z*) in peanuts were significantly up-regulated under salt stress (*Wanner et al., 1995*; *Zhang et al., 2020b*). *Singh, Singh & Singh (2013)* showed that cinnamic acid could reduce the accumulation of ROS in maize plants by providing ROS-scavenging enzymes under salt stress. We found that the expression levels of *PAL* family genes (*GmPAL1.2, GmPAL1.3, GmPAL2.2, GmPAL2.4*) were also significantly up-regulated in salt-treated soybeans (Fig. 4), indicating that these genes were involved in the response to salt stress. However, the means by which these genes exert their regulatory functions requires further study.

TFs are key regulatory factors in the complex signal network of plants in response to abiotic stress. The function of several TFs in the regulation of salt stress in plants has been reported, including that of AP2/ERF, MYB, bHLH, WRKY, and C2H2 (*Ciftci-Yilmaz et al., 2007*; *Li et al., 2019a*; *Liu et al., 2015*; *Makhloufi et al., 2014*; *Zhao et al., 2019b*). In this study, 116 TFs were found in the differential genes by RNA-seq, accounting for 9.4% of the DEGs. These TFs were distributed among different families, including AP2/ERF, bZIP, NAC, MYB, BHLH, C3H, C2H2, HD-ZIP, and WRKY, etc. The AP2/ERF family has the largest number of members (32), followed by MYB (17) and bHLH (14) (Fig. 5). This is
consistent with transcriptome analysis results from other species undergoing salt treatment. *Long et al. (2019)* found that the number of the AP2/ERF family members was largest among the DEGs treated with salt, accounting for 17.6%. Further analysis showed that MYB had the most members (11) in the up-regulated TFs (Fig. 5). *Munsif et al. (2020)* also found that the MYB family has the largest amount of up-regulated differentially expressed TFs in salt-treated isonuclear kenaf. *Zhang et al. (2020c)* showed that overexpression of *SlMYB102* enhanced salt tolerance of transgenic tomatoes by regulating a series of molecular and physiological processes. *OsMYB2*-overexpressing transgenic rice plants were more tolerant than wild-type to salt stress (*Yang, Dai & Zhang, 2012*). *Chen et al. (2019)* showed that MdMYB46 can enhance the salt tolerance of apples by activating stress-related genes (*MdABRE1A*, *MdDREB2a*, *MdRD22,* and *MdRD29A*). TaMYB86B affects wheat's salt tolerance by regulating ion homeostasis, maintaining osmotic balance, and reducing ROS levels (*Song et al., 2020b*). Phylogenetic analysis revealed that GmMYB46 (LOC100799410), MdMYB46, and TaMYB86B were highly homologous in the same clade (Fig. 6A). Analysis also showed that the expression level of the *GmMYB46* gene was the highest up-regulated expression among all differentially expressed MYBs (Fig. 6B). Therefore, we believe that it is worth exploring the function of *GmMYB46* under salt stress.

Further analysis revealed that the expression level of *GmMYB46* first increased and then decreased under the treatment of mannitol compared with salt treatment (Fig. 6C). Plants first go through a rapid osmotic stress phase when exposed to salt in the environment before experiencing ion toxicity caused by salt stress (*Zhao et al., 2019a*). Mannitol treatment led to osmotic stress. The expression pattern of *GmMYB46* suggesting that it may function in the early response to osmotic stress, and its longer response time to salt treatment indicated that it may play an important role in salt stress. The salt tolerance of *Arabidopsis* overexpressing *GmMYB46* is significantly enhanced at 5 d and 7 d after salt treatment (Fig. 7). These results implied that *GmMYB46* plays an important role in both short-term salt response and long-term salt stress tolerance. *Chen et al. (2019)* reported similar results and found that a short-term salt responsive gene, *MdMYB46*, in apples plays an important role in enhancing the long-term salt stress tolerance of transgenic apples. MDA content, as a product of membrane lipid peroxidation, reflects the degree of damage of plant cells due to abiotic stress, including salt stress (*Song et al., 2019*). We found that the MDA content of overexpressed lines under salt stress was significantly lower than that of WT (Fig. 7). The enhanced activity of SOD and POD helps plants reduce oxidative damage (*Zhang et al., 2020d*) and the proline content may reflect the stress resistance of plants. We found that the activity of SOD and POD, and the content of proline were significantly increased in overexpression lines compared with WT under salt stress (Fig. 7). In addition, the key genes for proline biosynthesis *P5CS1* (*Liu et al., 2015*), and antioxidant enzymes SOD and POD encoding genes, were significantly up-regulated in the overexpression lines after salt treatment compared with WT (Fig. 8). *NCED3*, as a key gene in ABA synthesis (*Van Zelm, Zhang & Testerink, 2020*), was significantly upregulated under salt treatment, and the upregulated ratio in the overexpressed line was significantly higher than that in WT. These results further suggest that the regulation of GmMYB46 in the salt stress response is complex, and the detailed molecular regulatory mechanism requires additional study.

## CONCLUSIONS

The shoots of control and salt-treated soybean seedlings were used for transcriptome sequencing to analyze their molecular mechanisms in response to salt stress. Transcriptome analysis revealed a total of 1,235 DEGs, of which 466 genes were up-regulated and 769 genes were down-regulated. The DEGs involved in important pathways were determined by KEGG enrichment, which included 'Plant hormone signal transduction', 'MAPK signaling pathway', 'Arginine and proline metabolism', and 'Phenylalanine metabolism'. We further analyzed the DEGs in these pathways to characterize their possible functions in responses to salt stress as well as TFs related to salt stress responses. The phylogenetic trees of differentially expressed MYBs induced by salt stress were constructed. The salt tolerance of transgenic *Arabidopsis* overexpressing *GmMYB46* was significantly enhanced, and GmMYB46 was found to activate the expression of salt stress response genes (*P5CS1, SOD, POD, NCED3*) in *Arabidopsis* under salt stress. This work will provide a basis for future studies to discover novel salt-tolerance genes, further investigate the molecular mechanism of salt tolerance in soybeans, and cultivate strong salt resistant varieties.

### Funding
This work was supported by the Natural Science Foundation of Hunan Province (2020JJ4030), the Innovation Platform and Talent Plan of Hunan Province (2020NK4222). The funders had no role in study design, data collection and analysis, decision to publish, or preparation of the manuscript.

### Grant Disclosures
The following grant information was disclosed by the authors:
The Natural Science Foundation of Hunan Province: 2020JJ4030.
The Innovation Platform and Talent Plan of Hunan Province: 2020NK4222.

### Competing Interests
The authors declare there are no competing interests.

### Author Contributions
- Xun Liu conceived and designed the experiments, performed the experiments, analyzed the data, prepared figures and/or tables, authored or reviewed drafts of the paper, and approved the final draft.
- Xinxia Yang conceived and designed the experiments, performed the experiments, prepared figures and/or tables, and approved the final draft.
- Bin Zhang conceived and designed the experiments, performed the experiments, authored or reviewed drafts of the paper, and approved the final draft.

### DNA Deposition
The following information was supplied regarding the deposition of DNA sequences:
All sequences are available at NCBI: PRJNA741583.

## Data Availability

The raw measurements are available in the Supplementary Files.

## Supplemental Information

Supplemental information for this article can be found online at http://dx.doi.org/10.7717/peerj.12492#supplemental-information.

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
