# Peer review of "Transcriptome analysis and functional identification of GmMYB46 in soybean seedlings under salt stress"

_PeerJ, doi:10.7717/peerj.12492_

## Round 0.1 · original submission · Major Revisions

After careful evaluation of the review reports, I feel to revise the article significantly. Please revise the manuscript according to the reviewer's comments and provided point-to-point response. Reviewer-3 has raised some valid questions on overall experiments. Please try to focus on the queries and answer them.

Reviewer 1 ·

Basic reporting

Lin et al. identified short-term salt responsive 1235 DEGs using RNA-seq and reported the important function of GmMYB46 in salt tolerance in Arabidopsis. This submission has certain significance for soybean salt stress tolerance. However, this submission was not well prepared, and there were many writing errors. There are more than ten errors in the abstract alone!
Basic reporting
1. There are many writing specifications and English grammatical errors in the whole article. I marked some errors in the review pdf I uploaded. I suggest that an English expert revise the full text again, which will significantly improve the quality of the article.
2. Requires proper explanation of osmotic stress and ion toxicity (line 28-29).
3. Please carefully revise the reference according to PeerJ's requirements. I have marked some obvious errors.

Experimental design

1. Short-term salt stress response and long-term salt stress tolerance are different molecular mechanisms and often have different molecular regulatory networks. In this submission, the author actually only identified soybean short-term salt stress response (only at 6h) DEGs with extremely loose DEG restrictions (p < 0.05; |Log2 fold change| > 1), not salt-tolerant DEGs. Fortunately, they claim that a short-term salt responsive DEG (GmMYB46) plays an important role in salt tolerance (at 5d and 7d). This implies that GmMYB46 plays an important role in both short-term salt response and long-term salt stress tolerance, and the author should discuss it because the number of similar genes is very small.

Validity of the findings

2. Extremely loose DEG restrictions (p < 0.05; |Log2 fold change| > 1), add DEGs results of p < 0.01; |Log2 fold change| > 1.5 in supplementary, please feedback the number of DEGs and whether GmMYB46 is in new DEGs? Is the main annotation path the same as it is now?

Additional comments

3. Has the author published the RNA-seq data? Please provide SRA number and add this in text!
I suggest that the author make major revisions based on the above questions.

Annotated reviews are not available for download in order to protect the identity of reviewers who chose to remain anonymous.

·

Basic reporting

1. The use of the grammar structure and vocabulary needs to be improved. “Salinity is a major abiotic stress that limits plant growth and crop productivity.” It is suggested to change “a major abiotic stress”to“one of the major abiotic stress.”
2. The name of the species should be italicized in the whole article. Like Arabidopsis and Agrobacterium. Please change to the correct font format.
3. The species of the soybean used in the experiment was not informed. Please add the name of the species.

Experimental design

no comment

Validity of the findings

Fig 6C: In this picture, the expression level of GmMYB46 was firstly increased and then decreased under the treatment of mannitol compared with salt treatment, please describe it in detail and tell the reasons.

Additional comments

The section of discussion should not only describe the experimental results, but put forward the differences and improvement of the experimental results with predecessors.

Reviewer 3 ·

Basic reporting

English is flawed and needs to be improved.

Experimental design

no comment

Validity of the findings

no comment

Additional comments

1. I have doubts about the author's analysis direction. According to the results of TFs, it is unwise for the author to select candidate genes based on the number of DE gene family members, ignoring the family expression abundance, in other words, why not choose AP2/ERF, bZIP, NAC?

2. In the KEGG enrichment analysis, the authors had screened several candidate genes, but some of them were up-regulated and some were down-regulated. This requires the authors to give more functional verification and discussion.

3. The author's discussion has no focus, and the genes are not systematically related. This point needs to be revised

4. In the results section, the author’s description of the results is too verbose and needs to be streamlined.

---

## Round 0.2 · Minor Revisions

Reviewer 2 has some minor concerns. Please address these concerns so that a final decision may be made.

Reviewer 1 ·

Basic reporting

The authors have satisfactorily addressed all comments, the revised manuscript is now suitable for publication in PeerJ.
with kind regards,

Experimental design

no comments

Validity of the findings

no comments

Additional comments

no comments

·

Basic reporting

The use of the English language needs to be improved.

Experimental design

It is better to prove overexpression of GmMYB46 enhances the salt tolerance of transgenic soybean.

Validity of the findings

It is suggessted to improve the sharpness of the Figure 1(A).

Additional comments

The part of discussion is not profound enough. It is suggested to add the innovation and improvement of this experiment and previous experiments.

---

## Round 0.3 · accepted · Accept

The paper is much improved.